# Design of a 2 × 4 Hybrid MMI-MZI Configuration with MMI Phase-Shifters

**DOI:** 10.3390/ma12091555

**Published:** 2019-05-12

**Authors:** Boris B. Niraula, Conrad Rizal

**Affiliations:** 1SeedNanoTech and Consulting, Brampton, ON L6Y 3J6, Canada; 2Department of Electrical & Computer Engineering, The University of British Columbia, Vancouver, BC V6T 1Z4, Canada; crizal@ece.ubc.ca

**Keywords:** multimode interferometer, MMI, MZI, thermo-optical switch, SOI technology

## Abstract

This paper reports design of a 2 × 4 hybrid multimode interferometer-Mach-zehnder interferometer (MMI-MZI) configuration consiting of compact thermo-optical switches on the silicon-on-insulator (SOI) platform. The device consists of two identical MMI slab waveguides as power splitters and couplers that are connected with two identical MMI-based phase shifters, and linear tapers at both ends of the MMIs to minimize the power coupling loss. A thin Al pad is used as a heating element and a trench is created around this pad to prevent heat from spreading, and to minimize loss. The calculated average thermo-optical switching power consumption, excess loss, and power imbalance are 1.4 mW, 0.9 dB, and 0.1 dB, respectively. The overall footprint of the device is 6 × 304 $\mu m^{2}$. The new heating method has advantages of compact size, ease of fabrication on SOI platform with the current CMOS technology, and offers low excess loss and power consumption as demanded by devices based on SOI technology. The device can act as two independent optical switches in one device.

## 1. Introduction

The consistent and continuous demand/growth of high-speed internet with a higher volume of data processing has prompted the need for high-speed non-blocking optical switches with less sensitivity to polarization and operational wavelength [1]. These functionalities need to accompany with low power consumption, low loss, small device footprint, and a higher number of ports, full operational bandwidth that depends on the number of output ports, and a low production cost.

The silicon-on-insulator (SOI) material platform supports the design of compact wave-guiding devices with reasonable fabrication tolerance and cost, and has attracted renewed research interest. The demand for high-speed internet and the vast amount of data transmission calls for a new generation of photonic switches. Interested readers in the comparison of technical characteristics of presently operational and proposed optical switches based on MMI waveguides are referred to [2].

Two widely known photonic waveguiding configurations are commonly used in realizing switching, modulating, and filtering functionalities. These include Mach-zehnder interferometry (MZI) [3] and multimodal coupler-based interferometry (MMI) [4]. MZI-based waveguiding configurations consist of directional couplers (DCs), tappers and nano-sized optical wires of different dimensions and geometry [5] and operate over wide/broad optical bandwidth, and show environmental stability.

Most of the work reported in literature on optical switches have employed directional couplers (DC) as the power splitter and combiners where high-speed operation is achieved by varying the coupling coefficient of the couplers. For high speed operation, the distance between the adjacent waveguides in the DC needs to be small. However, that creates higher mode conversion optical losses, which is undesirable. The overall footprint of the DC is also large. In addition, most nano-sized waveguides used in DC have small fabrication tolerance and therefore, a tight control of the fabrication process is necessary.

MMIs offer benefits over DCs due to their advantages of compactness, ease of fabrication, wide band operation, and large fabrication tolerance (up to ±50 nm can be achieved). Moreover, the silicon photonic platform enables the possibility of fabricating and integrating the optical device with the electronics on the same platform in the commentary metal-oxide-semiconductor (CMOS) environment. This enables large-scale integration of photonic devices at a lower cost and higher volume production using CMOS-compatible fabrication method. In this regard, a MMI power coupler/splitter plays an important role in the development of integrated photonics due to the above-mentioned merits.

MMI-based wave-guiding configurations consist of micro-sized slab-like couplers/decouplers with connectors of different shape and geometry [4]. The MMI couplers work on the principle of self-imaging effect [6]. In addition, MMI-based couplers/decouplers as switching configurations offer many advantages such as well-defined decoupling/coupling ratios, lower excess loss, low polarization sensitivity, better dimensional tolerance and/or ease of fabrication, ease of cascading them and integrating with the CMOS technology, and wavelength insensitive transmission spectra, and can support many ports. However, the coupling ratios of the MMI splitters/couplers are very limited if only one of the MMIs is used for operation. The fixed coupling/splitting ratios would allow only a limited applications in all-optical signal processing and optical networks. Therefore, it is highly desirable to implement the couplers with variable coupling ratios. This can be achieved using phase shifters, in this case, two MMI phase shifters are added to the MZI arms to control the propagation constant/phase of the propagating waves.

In this paper, we have theoretically demonstrated all MMI based 2 × 4 configuration in which two separate MMI decouplers/couplers (each with 6 μm × 140 μm) are connected via two MMI-based phase shifters (each with 2 μm × 8 μm) within a compact footprint using 4 μm long tapered waveguides. The most important feature of this device is that it can act as two independent optical switches in a single device where the coupling ratios of the device can be readily varied by using two independent phase shifters. The working principle of the device is verified using the transfer matrix method and FDTD simulation tools. As demonstrated earlier in Reference [7], an Al heating element is incorporated in the phase shifter arms to realize thermo-optical switching.

## 2. Design, Simulation, and Optimization

### 2.1. Multimode Interferometers

The fundamental switching element in this work is MMI based phase shifters and thermo-optic based heaters. The design of the 2 × 4 optical switch, considered in this paper is based on single MMI couplers/splitters and their cascaded configuration, the operating principle of which is based on self-imaging theory and total internal reflection [6,8,9,10,11,12,13,14,15,16,17]. The device consisted of two identical MMIs: MMI${}_{1}$ and MMI${}_{2}$, each with 140 μm × 6 μm dimension, connected via two smaller phase shifters MMI${}_{3}$ and MMI${}_{4}$, each with 16 ${\mu m}^{2}$, and a metal heating element incorporated in them. The MMI${}_{1}$, in this case, is designed to act as a power decoupler whereas MMI${}_{2}$ is designed as a power coupler.

Figure 1 presents an illustrative configuration of the device that contains a 3-dB input MMI decoupler/splitter, two phase shifters, one at each arm of the MMI, and a 3-dB output MMI coupler. The MMIs are designed in such a way that when the input signal is fed through a${}_{i1}$, the output power is equally split between a${}_{o1}$ and a${}_{o4}$. Likewise, when the signal is fed through the input port a${}_{i2}$, the output power is equally split between a${}_{o2}$ and a${}_{o3}$. That means each MMIs can act as two independently operating 3-dB splitters/couplers.

Based on self-imaging properties, a prototype MMI 3-dB power splitter/combiner is first designed. According to [6], the beat length, $L_{\pi }$ between two lowest order modes of an MMI is given by:(1)$$L_{\pi }=\frac{\pi }{\left(\beta_{o}-\beta_{1}\right)}$$
where, $\beta_{0}$ and $\beta_{1}$ denote the propagation constants of the fundamental and first mode, respectively.

As explained by Ulrich [17] and Soldano [13], self-imaging is achieved when the input field is reproduced as single or multiple images along the dimension of the MMI slab at a periodic interval. Since the light beam is confined in an MMI slab because of obeying total multiple reflections, the beam undergoes multiple periodically repeating self-interference patterns. For instance, self-generation of two interference patterns with equal output powers can be achieved using the relationship as [6]:(2)$$L_{MMI}=M\times 3\times \frac{L_{\pi }}{N}$$
where *M* and *N* are any positive integers without a common divisor, *N* is the number of self-replicating interferences, and *M* defines device length with various *N* [16]. The relationship in (Equation 2) suggests that a compact device is obtained for *M* = 1.

For the MMI with the effective width, W${}_{e}$ and effective refractive index, n${}_{eff}$, and when excited from either input or output ports, the L${}_{\pi }$ can also be given by:(3)$$L_{\pi }=\frac{{W_{e}}^{2}\times 4\times n_{eff}}{\left(3\times \lambda \right)}$$

That means, $L_{\pi }$ is related to the W${}_{e}$, λ, and $n_{eff}$.

For a lateral wave-guided structure, the $W_{e}$ is related to the geometric width of the device as [13]:(4)$$W_{e}=W_{MMI}+\frac{\lambda }{\pi }{\left(\frac{n_{core}}{n_{cld}}\right)}^{2}\frac{1}{\sqrt{\left({n_{core}}^{2}+{n_{cld}}^{2}\right)}}$$
where $n_{core}$ and $n_{cld}$ are the refractive indices of core and cladding layers, respectively. At one beat length and a half, the MMI produces a pair of identical images, making the waveguide a 3-dB coupler.

As shown in Figure 1, the device is designed to act as two independent optical switches with tapered input/output ports (the tapered section is shown in the inset at the middle top for clarity and the cross-sectional view of the waveguide is shown at the top right). Air, with a refractive index of 1.0 is considered as the upper cladding whereas SiO${}_{2}$, with a refractive index of 1.45 as the lower cladding.

The individual MMI device parameters were estimated using MATLAB mode solver [18] and FDTD simulation tool with TE polarized (s-polarized) light at λ = 1550 nm and these are given in Table 1.

### 2.2. Access Waveguides

To ensure single mode operation, geometrical parameters of the access waveguides were derived using MATLAB mode solver [18] for the TE polarized (s-polarized) light at λ = 1550 nm and FDTD simulation tool. The core thickness of 0.22 μm and access waveguide width of 0.5 μm at the input side of the tapered waveguide were chosen to achieve a single mode operation. All the calculated geometrical parameters of the access waveguides are given in Table 1.

### 2.3. Phase Shifters

In many circumstances, the phase shifting is an essential characteristic of the optical signal as it traverses through waveguides, including switches and filters. Not all kind of phase shifting is, however, useful for switching application. In most cases, the desired phase shift requires special device design consideration. In the present work, two 1 × 1 MMI devices (MMI${}_{3}$ and MMI${}_{4}$) are considered for phase shifting purpose. Symmetric interference theory was used to calculate the L${}_{MMI}$ to give a single image at the output. In the simulation, the length of the MMIs were varied from 0 to 10 μm to obtain corresponding MMI width for single mode operation. The width was varied from 1 to 2 μm for a 8 μm long MMI to avoid cross-talk between the adjacent waveguides. The chosen waveguide can support up to a maximum of 3 optical modes. These parameters were obtained from the FDTD simulation as well as MATLAB Mode solver [18]. The grid-size used during FDTD simulation was 5 nm. In analogy to reference [7], two mesh override regions were used. The calculated parameters of the phase shifter are given in Table 1.

### 2.4. Thermo-Optic Heaters

The change in refractive index along the length of an optical (Si) waveguide leads to a change in mode propagation constant of the optical signal. Refractive index of a Si waveguide can be tuned using two effects: free career plasma dispersion and thermo-optical effect [7]. The thermo-optical effect is relatively stronger as Si exhibits a strong thermo-optical coefficient of $\frac{dn}{dT}$ = −1.8 × 10${}^{-4}$ K${}_{1}$ at 300 K.

## 3. Principle of Operation of the Device

The operating principle of the thermo-optic switch is based on the change of mode propagation constant of the beam as it propagates through the phase-shifter/modulating arm with respect to the reference arm. As shown in Figure 2, the optical transfer function of this switch is governed by the multimode interference principle that is achieved by modulating the phases of the transmitted light. The device’s input signal can be switched to either output of the device when the phase difference, changes from 0 to π between the MMI arms. A thin Al pad is introduced in both the phase shifter arms, MMI${}_{3}$ and MMI${}_{4}$, with the aim of varying the coupling coefficient and thus the output powers through MMI${}_{2}$ (discussed more in Section 4.3). Either one or both phase shifters can be tuned depending on the application to realize the desired phase shift.

The phase difference (Δϕ) between the light propagating through the phase modulating and reference arms depends strongly on the effective refractive indices and dimension of the waveguide and is given as [7]:(5)$$\Delta \phi =\frac{2\pi }{\lambda }\times \left(\Delta n_{eff}\times L_{MMI}\right)$$
where λ is the operating wavelength, 1550 nm in this case, $\Delta n_{eff}$ is the change in effective refractive indices, and $L_{MMI}$ is the length of the modulating arm. Various parameters of the waveguides and heater element are given in Table 2.

For devices with equal length of the phase modulating/shifter and reference arms such as the one considered in this case, a phase shift of π can be introduced with the help of a heater in the modulating arm as the refractive index is temperature dependent as [7]:(6)$$\Delta \phi =\frac{2\pi }{\lambda }\times \frac{\delta n_{eff}}{\delta T}\times \Delta T\times L_{MMI}\left(1+\alpha_{L}\times \Delta T\right)$$
where, $\delta n_{eff}$/δT is the thermo-optic coefficient and $\alpha_{L}$ is the coefficient of thermal expansion and for Si, it is 3.6 × 10${}^{-6}$/°C. At a constant $\delta n_{eff}$/δT and λ, the Δϕ is a function of L${}_{MMI}$ and ΔT. L${}_{MMI}$ is the heating section of the modulating arm. Thermo-optical coefficient of Si is 1.86 × 10${}^{-4}$ K${}^{-1}$ at λ = 1550 nm. A device design based on this mathematical formalism acts as a thermo-optical switch and using this formula, the calculated temperature change, ΔT of 26.77 °C is required to achieve a phase shift of π for a 8 μm long modulating MMI waveguide.

An electro-thermal joule heating was considered using an Al pad as the electrode. Al was chosen as a heating element because of its large thermal conductivity of 204 W/(m·K) at room temperature. The electrode size regarding the heating area was chosen to be 1.4 μm × 8 μm to match the length of the modulating/phase shifter arm. The calculated switching voltage of 1.79 V corresponds to the average maximum heating temperature of 50–52 °C as a phase shift of π is introduced through the heating pad in the modulating arm with respect to the reference arm.

To minimize heat loss and spreading of heat to reference arm and to other parts, a trench is created around the modulating arm as described earlier [7,20,21]. The implementation of this design significantly helps prevent heat spreading and minimize loss by over 90% as the heat required to drive the switch is directly linked to the device size. The parameters used for the calculation are given in Table 2.

## 4. Results and Discussions

### 4.1. Field Distribution in the MMI Couplers /Decouplers

Figure 3 shows the field distribution profile obtained at λ = 1550 nm from FDTD simulation along the length of MMI${}_{1}$ (Note that MMI${}_{2}$ = MMI${}_{1}$) when the input is fed through (a) a${}_{i1}$ and (b) a${}_{i2}$ (FDTD simulation is a general method to solve Maxwell’s partial differential equations in the time domain).

In Figure 3a, the incident power is fed from port 1 ($a_{i1}$) (see Figure 1 and Figure 2 for the schematics). As shown in it, for the given configuration, the optical field is tightly confined/concentrated at the center of the waveguide. The field profile suggests that the optical field is not lost through the walls of the waveguide suggesting that some of the light reflected back to the center before reaching the wall. That means, we can reduce the MMI width further (also evident from the field profile). The optical beam emerges from the output ports of the MMI${}_{1}$ with a power splitting ratio of a${}_{o1}$/a${}_{o4}$ = 0.48/0.47, which closely agrees with our design consideration.

Figure 3b shows the field profile of the MMI${}_{2}$ (Note that MMI${}_{2}$ = MMI${}_{1}$) as the incident power is fed from input port 2 ($a_{i2}$) (again, see Figure 1 for schematics). In this case, the output power is split into the a${}_{o2}$/a${}_{o3}$ = 0.46/0.45 ratio. The power splitting ratio suggests that it varies slightly depending on whether the light is fed through $a_{i1}$ or $a_{i2}$. The optimized length of each of the MMI coupler is calculated to be L${}_{MMI1,2}$ = 140 μm.

For the device, the insertion loss (IL) is given by:(7)$$IL=-10\;log_{10}\frac{P_{out}}{P_{in}}$$
and, the excess loss (EL) is given by:(8)$$EL=-10\;log_{10}\frac{P_{min}}{P_{max}}$$

For both the MMIs, the insertion loss and excess losses estimated using FDTD simulation came out to be around 0.65 dB and 0.09 dB, respectively. Note that the L${}_{MMI}$ of each MMIs (MMI${}_{1}$ and MMI${}_{2}$) was optimized to be 140 μm to achieve this ratio of power splitting, as this ratio and L${}_{MMI}$ are directly related to W${}_{MMI}$ and *n*, as shown in (Equation 3).

By connecting MMI${}_{1}$ and MMI${}_{2}$ (each with a length of L${}_{MMI}$ = 3 × $L_{\pi }$/2) as shown in Figure 3 together with the MMI phase shifters (MMI${}_{3}$ and MMI${}_{4}$) in the linking arms of the MZI configuration as shown in Figure 1, two independently operating optical switches (also shown in Figure 2) can be achieved. Both the phase shifters allow the coupling coefficients of the MMI${}_{2}$ to vary and tune.

The cross talk (CT), i.e., power coupling between the adjacent waveguides is given by:(9)$$Crosstalk=-10\;log_{10}\frac{P_{bar}}{P_{cross}}$$

The P${}_{bar}$ and P${}_{cross}$ in (Equation 9) denote a${}_{o11}$ and a${}_{o44}$, when the input is fed from a${}_{i1}$ and a${}_{o22}$ and a${}_{o33}$, when the input is fed from a${}_{i2}$, respectively. The width of the MMIs and linear access waveguides were chosen to be 6 and 0.48 to 0.8 μm, respectively to improve the device performance.

### 4.2. Field Distribution in the MMI Phase Shifters

Since the design and realization of the thermo-optical switch requires several practical considerations, the optical field and mode propagation profiles of the phase shifter MMIs are studied as a first step.

The optical field profile of MMI${}_{3}$ is shown in Figure 4a. The profile suggests that the optical wave is tightly confined in the center of the waveguide with single-mode propagation and with no noticeable energy loss through side walls. This field profile also suggests that the width of this waveguide can be reduced by up to 10% without sacrificing the operation (the video showing the propagation of field inside the phase shifter is given in the Appendix A).

The intensity profile/power profile of the phase-shifter arm as a function of the MMI${}_{3}$ is shown in Figure 4b. As shown in it, the power coupling ratio can be tuned/changed by varying the phase shift of the phase modulating arm (i.e., L${}_{MMI_{3}}$). The output to input power ratio is found to be 0.9 to 1.0 (the y-axis label is not shown here), which means, there is a 10% loss of power. The output power gradually builds up and at the exit, it goes of up to 90% (Note in Figure 4b the magnitude of the field intensities at $L_{MMI}$ = 8 and 16 μm).

### 4.3. Optical Characteristics of the Switching Device

Figure 5 shows the schematics of the proposed device (cascaded MMI couplers) and the optical switch parameters (expressed in the normalized unit) used for calculation and run FDTD simulations. Each of the optical switches shown here operate in a binary state (1,0), depending on whether the output power state is in the bar or in the cross ports of MMI${}_{2}$.

#### 4.3.1. Case I: Δϕ= 0, 0 (Heating Pads in MMI${}_{3}$ are Disabled)

For the light input at port a${}_{i1}$, preliminary simulation results suggested that the normalized output powers from port 1 to 4 are approximately equal to 0, 0, 0, 0.75, respectively when phase shifting is not considered (i.e., $\Delta \phi_{MMI3}$ = $\Delta \phi_{MMI4}$ = 0). The excess loss and power imbalance estimated using (Equation 7) and (Equation 8) are found to be 0.95 dB and 0.1 dB, respectively.

#### 4.3.2. Case II: Δϕ= π/2, 0 (Heating Pads in MMI${}_{3}$ are Enabled)

Upon the introduction of π/2 phase shift in the first phase shifter, MMI${}_{3}$ (i.e., $\Delta \phi_{MMI3}$ = π/2 and $\Delta \phi_{MMI4}$ = 0), the normalized output powers at the output ports are estimated to be ≈ 0.48, 0, 0, 0.46, respectively. The excess loss (Equation 7) and power imbalances (Equation 8) estimated from the FDTD simulation are 0.90 dB and 0.10 dB, respectively.

#### 4.3.3. Case III: Δϕ= 0.85π, 0.40π (Heating Pads in MMI${}_{3}$ are Enabled)

Upon the introduction of 0.85π phase shift in the first phase shifter, i.e., MMI${}_{3}$, and 0.40π in the second phase shifter MMI${}_{4}$ (i.e., $\Delta \phi_{MMI3}$ = 0.85π and $\Delta \phi_{MMI4}$ = 0.4π ), the normalized powers at the output ports are estimated to be ≈ 0.75, 0, 0, 0.10, respectively.

When the heating pad is enabled (Case III), the cross-talk of the device when the input is fed from a${}_{i1}$ is found to be ≈−35 dB (output is taken from a${}_{o11}$ and a${}_{o44}$). Table 3 lists various switching performance characteristics (output power state for three different phase modulating cases) of the optical switch designed in this work. It shows that the device can be used as multipurpose optical switch. If the phase shifters operate fast enough, then the device may be a very promising building block for use in many types of optical devices.

We also studied the effect of phase shift on output powers (a${}_{o22}$ and a${}_{o33}$) using the phase shifter MMI${}_{4}$ arm (not shown here). The cross-talk of the device when the input is fed from a${}_{i2}$ is found to be ≈−34 dB.

### 4.4. Output Power State vs Input Driving Power

Switching characteristics of our optimized MMI design are shown in Figure 6 for the signal fed from a${}_{i1}$ and then a${}_{o1}$ (input of MMI${}_{3}$). It shows a plot of output power state as a function of heating power, changed in the range of 0 to 2.4 mW. At 1.4 mW and using the resistivity of Si (2.3 × 10 ${}^{3}$
μm), the voltage required to switch the state from 1 to 0 is calculated to be around 1.79 V. The reduction in power loss is almost by a factor of 2 as compared to the conventional phase shifter without the trenches [22,23]. The results suggested that the total device length should be chosen to be 304 μm to achieve the best operation. These results are comparable and consistent with the results estimated using self-imaging theory [6,24].

Device dimension and fabrication tolerances have a significant effect on the cross-talk of the optical switches. For example, higher the variation on device dimension and phase shifts, higher the degradation of the cross talk can occur. Proper control of phase shifting in MMI couplers, and tight control of fabrication parameters are needed to improve the cross-talk of the device below −35 dB. The requirements for idealized switching using thermo-optic phase shifters is even more stringent than the switching without the thermo-optic heaters. For example, a small variation of the dimension of the fabricated heater element can degrade the cross talk significantly. For multipurpose switching operation, the phase-shifting and tuning of the transmitted power can be achieved by placing heater elements in any of the four linking arms of the MMIs discussed in this paper.

## 5. Conclusions

In this paper, we theoretically demonstrated the operation of 2 × 4 multimode interference optical switch with a foot print of 304 μm${}^{2}$ on SOI technology. The device acts as two independently operating digital switches with improved efficiency and lower footprint, depending on the amount of phase shift introduced by the thermo-optic heaters. As obtained by simulation, the device showed a small insertion loss of around 0.9 dB, excess loss in the range of 0.2 to 0.7 dB, and cross-talk of around −35 dB. Creating a trench around the Al heating pad and isolating the heater from the surrounding media, the device required a small heating power of 1.4 mW and a switching voltage of around 1.79 V [7] to modulate phase shift from 0 to π. The new heating method offers the possibility of designing low excess loss and low power consumption optical devices on SOI technology. In addition to using as optical switches, these MMI-based devices have a potential to be used for optical interconnects and micro-ring resonator applications. Moreover, the work can be extended to biosensing that can be used to detect various media such as poisonous gas, environmental changes, as well as diseases caused by harmful bacteria and viruses.

## Figures and Tables

**Figure 1 materials-12-01555-f001:**
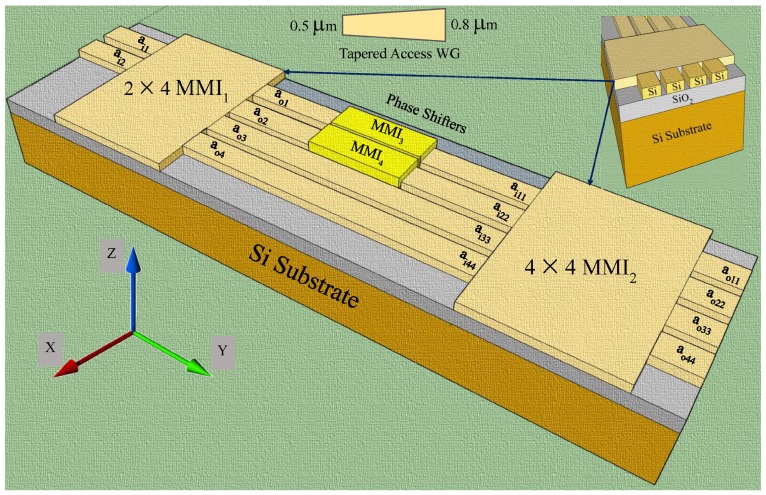
Schematics of the MMI device: Decoupler (MMI${}_{1}$) and Coupler (MMI${}_{2}$). The inset in the top middle shows a tapered access waveguide and in the top right the cross-sectional view of the MMIs. Input ports: a${}_{i1}$ and a${}_{i2}$ of MMI${}_{1}$ and output ports: a${}_{o11}$, a${}_{o22}$, a${}_{o33}$, and a${}_{o44}$ of MMI${}_{2}$.

**Figure 2 materials-12-01555-f002:**
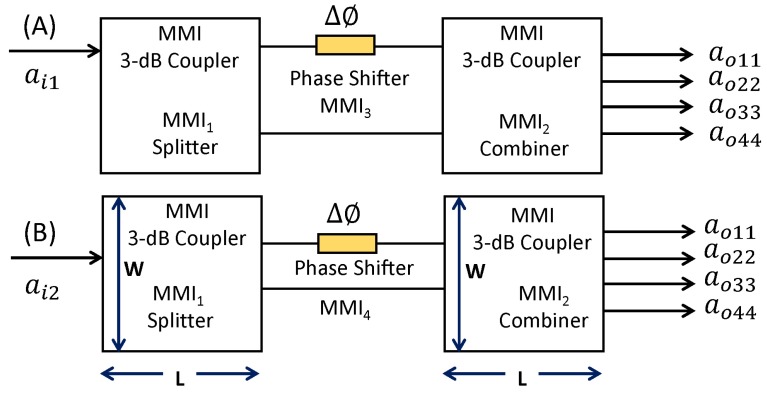
The schematics showing the principle of operation of optical switches. It shows the top view of the MMI${}_{1}$/MMI${}_{2}$, and phase shifters MMI${}_{3}$/MMI${}_{4}$, and corresponding waveguide width, length, and positions of access waveguides. The device parameters were calculated using MATLAB Mode Solver [18] as well as FDTD simulation [19]. The device acts as two independent optical switches depending on whether the input is fed through a${}_{i1}$ (**A**) or a${}_{i2}$ (**B**).

**Figure 3 materials-12-01555-f003:**
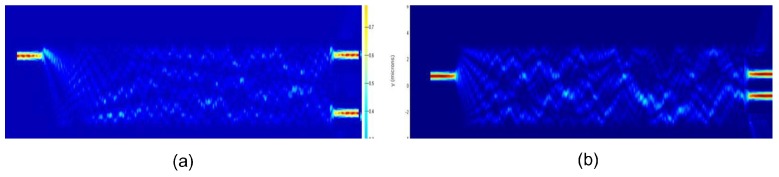
Electric field inside the the access waveguides and MMI${}_{1}$ where the input is fed through (**a**) port a${}_{i1}$ and (**b**) port a${}_{i2}$. Two-dimensional FDTD numerical method was used to optimize and verify the design [19], as it was shown to produce sufficiently accurate results in simulating devices based on SOI channel waveguides.

**Figure 4 materials-12-01555-f004:**
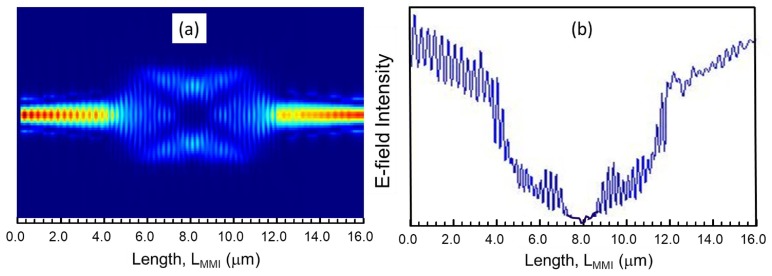
(**a**) Field profile of a MMI${}_{3}$ phase shifter without a heater. The 4 μm long tapered access waveguides are connected at both ends of the MMIs to reduce power losses (**b**) Intensity profile of the 1 × 1 MMI${}_{3}$ phase shifter as a function of L${}_{MMI_{3}}$.

**Figure 5 materials-12-01555-f005:**
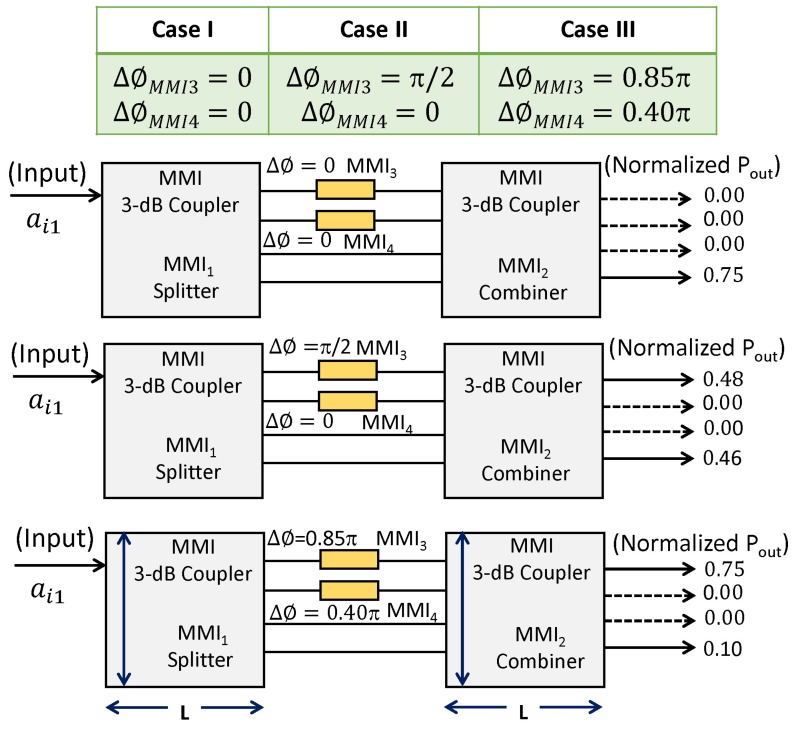
Cascaded MMI devices as tunable optical switches. Case I: The input is fed through a${}_{i1}$
$\Delta \phi_{MMI3}$ = $\Delta \phi_{MMI4}$ = 0, Case II: $\Delta \phi_{MMI3}$ = π/2 and $\Delta \phi_{MMI4}$ = 0, and Case III): $\Delta \phi_{MMI3}$ = 0.85 π and $\Delta \phi_{MMI4}$ = 0.4 π. The dark bold arrows indicate the light input and output positions and the numbers are the normalized optical powers.

**Figure 6 materials-12-01555-f006:**
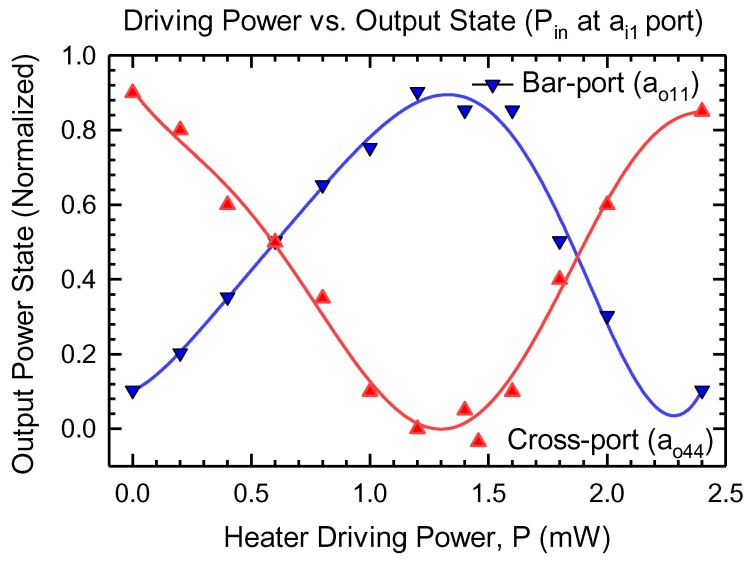
Calculated powers from the output ports of the MMI${}_{2}$: bar (a${}_{o11}$) and cross (a${}_{o44}$) as a function of the heating power in MMI${}_{3}$, and input is fed at a${}_{i1}$. The solid lines are fitted curve using KyPlot [25].

**Table 1 materials-12-01555-t001:** Geometrical parameters of the 3-dB splitters/couplers, access waveguides (AWs), and phase shifters (PS) obtained from simulation. The symbols W, L, and t denote width, length, and thickness of the device, respectively. Optical parameters used for simulation include substrate, Si (n = 3.48), lower cladding, SiO${}_{2}$ (n = 1.45), core, Si (n = 3.48), and upper cladding, air (n = 1.0).

Components	Width (μm)	Length (μm)	Core Thickness (μm)
MMIs: MMI${}_{1}$, MMI${}_{2}$	W${}_{MMI}$ = 6.0	L${}_{MMI}$ = 140	t${}_{MMI}$ = 0.22
Access Waveguides, AWs	W${}_{AW}$ = 0.5–8.0	L${}_{AW}$ = 4	t${}_{AW}$ = 0.22
PSs: MMI${}_{3}$, MMI${}_{4}$	W${}_{PS}$ = 1–2	L${}_{PS}$ = 8	t${}_{PS}$ = 0.22

**Table 2 materials-12-01555-t002:** Physical properties of the waveguide and heater element.

SN	Material	Thermal Conductivity, W/(m·K)	Heating Coefficient (/K)
1	Si	163.25	160 × 10${}^{-6}$
2	SiO${}_{2}$	1.405	8 × 10${}^{-6}$
3	Al	204	24 × 10${}^{-6}$

**Table 3 materials-12-01555-t003:** Switching state of the device. The signal is fed at the input port a${}_{i1}$.

Case	ΔΦ	Output Position	Output State (Normalized Power)
I	0, 0	a${}_{o11}$, a${}_{o12}$, a${}_{o13}$, a${}_{o14}$	0,0,0,0.75
II	π/2, 0	a${}_{o11}$, a${}_{o12}$, a${}_{o13}$, a${}_{o14}$	0.48,0,0,0.46
III	0.85π, 0.40π	a${}_{o11}$, a${}_{o12}$, a${}_{o13}$, a${}_{o14}$	0.75,0,0,0.10

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
