# Peer review of "Design of a 2 × 4 Hybrid MMI-MZI Configuration with MMI Phase-Shifters"

_materials, 2019, doi:10.3390/ma12091555_

Reviewer 1 Report

See attached report

Reviewer 1 Report

We thank Reviewer 1 for providing valuable feedback and comments to improve the quality of our paper. Our response is given in bold red font.

Summary:

Reviewer 1: This manuscript describes a multimodal mode coupler using thermo-optical phase shifters. The geometry of the MMI is provided as well as some electromagnetic mode simulation.

Authors should clearly define the objective of the device. Is this a 2x4 switch, a mode converter, a multiplexer, between which ports? Once this is established, authors should provide the EM simulation and characteristics metric of the device for ALL possible configurations: loss, efficiency, cross coupling.

Response: We apologize for the confusion. We thank Reviewer 1 for proving this valuable comment. The device is designed to act as two independent optical switches. We have revised our title, abstract, results and conclusion section to clarify and address this.

We have added Fig 2 in the revised paper to address the concern. We have also added a paragraph in the revised text as: “The device acts as two separate optical switches in one structure. Almost most of the work reported in the literature on optical switches have employed directional couplers (DC) as the power splitter and combiners where the high-speed operation is achieved by varying the coupling coefficient of the couplers. For high-speed operation, the distance between the adjacent waveguides in the directional coupler needs to be small. However, that creates higher mode conversion optical losses, which is undesirable. The footprint of the DC is large. Also, most nano-sized waveguides used in DC have small fabrication tolerance. MMIs offer benefits over DCs due to their advantages of compactness, ease of fabrication, large fabrication tolerance. Further, the silicon photonic platform enables the possibility of integrating the optical device with the electronics on the same platform in the commentary metal-oxide-semiconductor (CMOS) environment.”

We have provided additional simulation results and characteristics metric of the device. A paragraph is added in Section 4: Results and Discussion, in subsection 4.1 as: By connecting MMI1 and MMI2 (each MMI with a length of LMMI = 3 x Lp/2) shown in Fig. 3 together with the MMI phase shifters (MMI3 and MMI4) in the linking arm of the MZI configuration as shown in Fig.1, two-independent optical switches, as shown in Fig. 2, can be achieved. Both the phase shifters allow the coupling coefficients of the MMI2 to vary and tune.

We have also added Fig. 5. Only the results obtained from the FDTD simulation carried out for the entire device (MMI1, MMI2, MMI3, and MMI4) are mimicked by the schematics. We hope this clarifies. We thank the reviewer for this critical feedback.

Reviewer 1: The speed of the device is not established, so the title could not claim “high-speed”.

Response: We thank the reviewer 1 for pointing out this error. We agree with the Reviewer that the speed information was not included in the first draft, and we are sorry for it. We have revised the title. The speed of the switch would depend on the ability of the phase shifters to operate fast enough, in that case, the device could be a very promising building block for optical switches, optical modulators, and optical add-drop multiplexers. We have addressed this in the revised text.

Reviewer 1: Acronyms such as MMI, IL, EL should be defined the first time it is encountered.

Response: We have defined the acronym in the revised text starting from the abstract.

Reviewer 1: Eq.1: missing denominator parenthesis

Response:  A right parenthesis in the denominator in Eq 1 is included in the revised text.

Reviewer 1: I do not see any use for figure 2 — length vs width. A 3D graph length/width/efficiency would be much preferable.

Response: We thank the reviewer for this comment. We agree that figure 2 is unnecessary. We have revised the text to include the information about the phase shifters and have removed original Figure 2 and replaced it with a new Figure. We are sorry that due to our time limit, we are not able to carry out further simulation of the device, but we would be mindful of creating a graph that would provide length/width/efficiency information in our future works.

Reviewer 1: Thermal dilatation coefficient should be introduced next to the thermo-optical coefficient (page 4). Table 2 should include thermal dilatation coefficient and thermo-optical coefficient.

Response: We thank Reviewer 1 for this valuable suggestion. Equation (6) is updated as in Ref [7], our earlier work on thermo-optic heaters. It was a typo on our part. The thermal expansion coefficient is is now included in (6), and the device parameters are calculated accordingly. As suggested, the this information is updated in revised Table 2.

Reviewer 1: I do not understand the sentence: “Note that the length of the modulating arm needs to be doubled to achieve a phase difference of π/2.” It seems to me that it should be half.

Response: Looking at the phase shift vs length of the waveguide for a fixed MMI width, indeed the length needs to be cut into half to achieve the π/2 phase shift. The sentence is removed from this position as it is not directly linked to the results presented in the text.

Reviewer 1: Authors claimed twice that the width of the device could be reduced by at least 20%, but do not provide any evidence that this reduction will not impact the performance of the device.

Response: As shown by the simulation in Fig. 3(a), the electric field is concentrated in the center and reflected from the wall of the MMIs. Not all the light reached to the wall of the MMI is reflected to the center. From this observation, we can estimate that the waveguide width can be reduced by about 10-20% and still achieve the desired splitting/coupling ratio. Sorry, this is not the most accurate method to estimate the MMI waveguide width, however, on closely observing the field profile, we can see that a room exist to reduce the MMI width which we can confirm from the further simulation.

Reviewer 1: Figure 3 and associated text: it is confusing that panel b is said to be MMI_2. This should be MMI1 both cases.

Response: We thank the Reviewer. Yes, we agree. While the geometrical parameters are the same for both the MMIs the function is different. MMI1 works as a power splitter where MMI2 as a coupler. This information is updated in the revised text. See for example, in Section 2.1 and Section 4.1 we have provided a Note as:  “Note that MMI1=MMI2,. Hope this clarifies.

Page 6, authors wrote: “The output to input power ratio is found to be 0.9 to 1.0, which means, there is a 10 % loss of power. This value suggests that there is destructive interference in the middle of the phase shifter.” I do not see the connection between these two phenomena.

Response: We are talking about the phase shifter arm here. Our simulation results showed that for the input of 1 mW optical power, output power from the phase shifter without the phase shifting is 0.9 mW. The power loss with in the MMI is most probably due to the destructive interference of the optical light. To avoid confusions, we have revised the text.

Reviewer 1: Figure 4a: there is no scale for the micron unit.

Response: We thank the reviewer for noticing this error. The scale is corrected in the revised text.

Reviewer 1: Figure 4b: what is the negative length? It would be much better to plot the coupling intensity according to the phase shift/temperature as explained in the text.

Response: The negative sign indicates the negative axis from the origin, chosen during the simulation. The diagram is appropriately revised. We have revised the caption to address this concern.

Authors explain that: “the power coupling ratio can be tuned/changed by varying the phase shift (i.e., L MMI3 ) of the phase modulating arm.” That would be a problem for the overall transmission of the system. The MMI3 should only change the phase, not the transmission.

Response: Yes we agree with the Reviewer. The MMI3 only changes the phase of the optical radiation by tuning the temperature. The output from MMI3 is fed to the input of MMI2  (ai14). The signal input from ai44 and ai14 then interact inside MMI2 and the merge in the output of ao14 or ao44 depending on the phase shift introduced in MMI3. We hope this clarifies.

Reviewer 1:  Section 4.3.1. And 4.3.2: “the normalized output powers from port 1 to 4 are 0, 0, 0, 0.75”. Authors should provide higher precision, eventually expressed in dB. It is important to understand the cross coupling between the ports. The standard for commercial MMI is -39dB. There should be 4 cases: heater 1 on/off, heater 2 on/off.

Response: We thank the reviewer for these crucial suggestions. In this paper, we have shown the case for Heater 1 On/Off only for simplicity (New Figure 5 is added to demonstrate the switching functionality in two cases Dj1=Dj2=0 and Dj1= p/2 and Dj2=0). For simplicity, the output power is normalized between 0 and 1. We hope that the new diagram (Fig 5) would clarify the functionality of the optical switch and assist the general reader in understanding the functionality of the optical switch.

Reviewer 1: “The power transmission coefficients of both these couplers can be adjusted using the phase shifter arms MMI 3 and MMI 4. How could the power transmission of MMI1 be impacted by downstream elements?

Response: We thank Reviewer 1 for this important comment. Indeed, the power transmission of MMI1 is not affected by the downstream elements. We were talking about the overall power transmission for the device. We have revised the text to address the concern and hope this clarifies.

Reviewer 1: “the device is energetically viable” meaning?

Response: We wanted to state that the device consumes less power/energy compared to similar devices reported in the literature. We have revised the text to address the concern.

Reviewer 1: “The results suggested that the total device length should be chosen to be 324 μm to achieve the best operation” The results suggest the size of the different individual elements, but not the overall size that include connection arms.

Response: The length 304 μm (Not 324 as reported earlier) is the total device length:

140 μm (MMI1)+140 μm (MMI2)+16 μm (MMI3) + 4 μm (Input Taper) + 4 μm (Output Taper)=304 μm.

So the total footprint of this device is 304 x 6 μm2. We hope this clarifies.

Reviewer 1: Figure 5: axis label “drivng” driving

Response: We thank the reviewer for noticing this error. We have corrected the typo in the revised diagram.

Reviewer 1: “The proposed design was fabricated at IMEC Belgium, but the measurement could not be carried out due to the unavoidable circumstances.” This is weird.

Response: We thank the reviewer for this feedback. Since the paper reports only theoretical/simulated results, the sentence is removed from the text.

Reviewer 2 Report

In this paper, 2x4 hybrd MMI-MZI optical switch is investigated.

However, authors should explain device operation principle in more detail.

1) The authors claim this switch realize high speed operation.

   However, switching speed does not seem to be discussed.

2) MMIs with three modes are used as a phase shifter. 

  However, the reason why MMIs are employed is not enough explained.

  Other types of phase shifter may be available. 

  The superiority of using MMI should be explained.

3) In section 4.3.2, the port a_i2 is used as an input port.

  In that case, the output power of MMI_1 is launched into only a_i2 and a_i3, according to Fig. 3(b).

  Then, port a_i1, and a_i4 seem to be not required.

  However, the output power of MMI_2 is obtained from port a_i1 and a_i4.

4) In this paper, the splitting ratio of 0.48/0.46 and 0.75/0.25 is realized.

  However, the motivation of realizing this splitting ratio does not seem to be explained.

5) In Fig. 5, bar-port and cross-port denote port a_i1 and a_i4 ?

   What do mark and line respectively means ? The marks is not on the line.

Reviewer 2 Report

We thank Reviewer 2 for providing valuable feedback for our paper. Our response is given in bold red font.

Reviewer: Comments and Suggestions for Authors: In this paper, a 2x4 hybrid MMI-MZI optical switch is investigated. However, authors should explain the device operation principle in more detail.

Response: We thank Reviewer 2 for the feedback. To avoid confusions, we have revised our title that reflects the results reported in this paper. To address, the device operation principle is also added in the text with more details. To address the concern, the principle of operation of the device, a new section is created by merging section 2.1 and section 3.4 (in the original paper) and placed under Section 3 “Principle of Operation” heading.

Reviewer: 1) The authors claim this switch realize high-speed operation. However, switching speed does not seem to be discussed.

Response: We thank Reviewer 2 for providing this important feedback, which we missed to address in the first draft. Our title has been revised to address this concern. In the revised text, we proposed that this device speed can be significantly improved.

Reviewer: 2) MMIs with three modes are used as a phase shifter. However, the reason why MMIs are employed is not enough explained. Other types of phase shifter may be available. The superiority of using MMI should be explained.

Response: We again thank Reviewer 2 for the comment. We have added a paragraph in the introduction section to address the concern as: “The coupling ratios of the MMI splitters/couplers are very limited to the fixed coupling ratios if only one of the MMIs is used for the operation. Therefore, it is highly desired to implement the couplers with variable coupling ratios. In this work, two MMI phase shifters are added to the MZI arms to control the phase of the propagation. MMIs are chosen over the DCs due to the ease of fabrication, large fabrication tolerance and ease of cascading them and integration with the CMOS technology.” To improve the length of conventional MZI switch design, a novel shorter SOI MMI coupler is proposed for light splitting and combining in the Mach–Zehnder interferometer (MZI) switch. The revised text is denoted by red.

Reviewer: 3) In section 4.3.2, the port ai2 is used as an input port. In that case, the output power of MMI1 is launched into only ai2 and ai3, according to Fig. 3(b). Then, port ai1 and ai4 seem to be not required. However, the output power of MMI2 is obtained from port ai1 and ai4.

Response: Yes, we agree. The MMIs are so designed to work as 3-dB coupler that when the input is fed through ai1, the output is obtained from ao1 and ao4. Likewise, when the input is fed through ai2, as you pointed, the output is from a02 and ao3. This way the same device can be used as two independent optical switches. These points are made clear in the new figure we added in the text (Fig 2 in the revised paper). We hope this clarifies.

Reviewer: 4) In this paper, the splitting ratio of 0.48/0.46 and 0.75/0.25 is realized. However, the motivation for realizing this splitting ratio does not seem to be explained.

Response: The MMIs shown in Fig 1 are designed to act as 3-dB coupler/splitters. So, the output power ratio in the range of 0.48/46 are close enough to confirm that the power output from the first MMI1 is indeed split almost in half. By modulating the phase shifter, we can control the output power in various ratios, which is one of the most important features of this device unlike other optical switches where the ratio of output is fixed this one allows to vary the output ratio of MMI2 device by modulating the phase in the phase shifter arm. We hope this clarifies.

Reviewer: 5) In Fig. 5, bar-port and cross-port denote port ai1 and ai4? What do mark, and line respectively means?

Response: Calculated output power from the output ports of the MMI2: bar (ao11) and cross (ao14) as a function of the heating power in MMI3, and input is fed at ai14. The solid line is the fitted curve with smooth line fitting using the KyPlot software, as given by Ref [24} in the text. The triangular symbols are the calculated power corresponding the heating power. Hope this clarifies.

We thank the reviewer.

Round  2

Reviewer 2 Report

The authors extensively revised their manuscript. Thus, I recommend this paper to be published.

Materials EISSN 1996-1944 Published by MDPI AG, Basel, Switzerland RSS E-Mail Table of Contents Alert
Back to Top